# Caregiver experiences of an integrative patient-centered digital health application for pediatric type 1 diabetes care: Findings from a pilot clinical trial

Shazhan Amed[1,2]*, Susan Pinkney[1,2], Fatema S. Abdulhussein[1], Anila Virani[3], Carlie Zachariuk[1,2], Sukhpreet K. Tamana[1,2], Shruti Muralidharan[1,2], Matthias Görges[2,4], Bonnie Barrett[2], Tibor van Rooij[2,5], Elizabeth M. Borycki[6], Andre Kushniruk[6], Holly Longstaff[7], Alice Virani[8], Wyeth W. Wasserman[2,8], on behalf of the TrustSphere Collaborative

1 Department of Pediatrics, University of British Columbia, Vancouver, British Columbia, Canada, 2 BC Children's Hospital Research Institute, Vancouver, British Columbia, Canada, 3 Population Health and Aging Rural Research Centre, School of Nursing, Thompson Rivers University, Kamloops, British Columbia, Canada, 4 Department of Anesthesiology, Pharmacology & Therapeutics, University of British Columbia, Vancouver, British Columbia, Canada, 5 Department of Computer Science, University of British Columbia, Vancouver, British Columbia, Canada, 6 School of Health Information Science, University of Victoria, Victoria, British Columbia, Canada, 7 Director Research Integration and Innovation, Provincial Health Services Authority, Vancouver, British Columbia, Canada, 8 Department of Medical Genetics, University of British Columbia, Vancouver, British Columbia, Canada

* Shazhan.amed@ubc.ca

## Abstract

Diabetes technology generates vital health data, but healthcare professionals (HCP) and patients must navigate multiple platforms to access it. We developed a digital health platform, co-designed with patients and families living with type 1 diabetes (T1D) and their HCPs, that aim to support a collaborative care experience through shared access to diabetes data, clinical recommendations, and resources. We describe caregivers' views on the platform's impact on clinic visits and child self-management in children with T1D. A six-month observational pilot study at BC Children's Hospital Diabetes Clinic in British Columbia, Canada, gathered data through surveys and interviews. Surveys were administered to caregivers and HCPs at different time points throughout the study; 18 qualitative interviews were conducted with caregivers at the conclusion of the study. Quantitative data were summarized descriptively. Interview data were transcribed, coded using open and systematic coding, and subsequent inductive thematic analysis. Eighteen caregivers completed the surveys, and 11 HCP participants submitted 41 surveys (approximately 3–4 each) after using the platform. Most caregivers (61%; 11/18) found the platform helpful, and 56% (10/18) reported that using the platform made their clinical visits and recommendations more personalized. Nearly all HCPs (90%; 37/41) were satisfied with the platform's ability to support clinical visits. Themes identified from caregiver

**Data availability statement:** All relevant data are within the manuscript and its Supporting information files.

**Funding:** This study was funded by the University of British Columbia's Office of the Vice President of Research and Innovation (https://research.ubc.ca) [Award Number: AWD-015766 to SA], in collaboration with Canada's Digital Technology Supercluster (https://digitalsupercluster.ca/) [Award Number: 2020_3 TRUSTSPHERE to TrustSphere Consortium], MITACS Inc. (https://www.mitacs.ca/) [Award Number: AWD-017050 to SA, WWW, MG, AV], and industry partners including IDENTOS (https://www.identos.com/), Careteam Technologies (https://www.getcareteam.com/), and Smile Digital Health (https://www.smiledigitalhealth.com/). The funders had no role in study design, data collection and analysis, decision to publish, or preparation of the manuscript.

**Competing interests:** I have read the journal's policy and the authors of this manuscript have the following competing interests: Shazhan Amed holds a BC Children's Hospital Research Institute Salary Award. SA is the Founder and CEO of Haibu Health, which is commercializing the digital platform described, and may benefit financially. SA has participated on advisory boards for Dexcom, Abbott, Novo Nordisk, Eli Lilly, Sanofi, and Insulet, unrelated to this work. Matthias Görges holds a Michael Smith Health Research BC scholar award (SCH-2020-0494). Elizabeth M Borycki holds a Michael Smith Health Research BC Health Professional Investigator award. Susan Pinkney, Fatema S Abdulhussein, Anila Virani, Carlie Zachariuk, Sukhpreet K Tamana, Shruti Muralidharan, Bonnie Barrett, Tibor van Rooij, Andre Kushniruk, Holly Longstaff, Alice Virani, Wyeth W Wasserman report no conflicts of interest.

qualitative interviews revealed that (1) the platform provided a convenient connection that improved preparedness and empowered caregivers in managing their child's T1D; (2) the platform's value was driven by the healthcare team's usage of it; and (3) caregivers felt hopeful that the platform could better support their child's T1D management. The platform could foster a collaborative and personalized care experience that enables caregivers to engage in diabetes self-management and feel connected to their healthcare team. These results will guide the future development, evaluation, and implementation of the platform.

## Author summary

Managing type 1 diabetes (T1D) involves keeping track of a lot of health information, like blood sugar levels, insulin doses, and food intake. Right now, families and healthcare providers often need to use several different apps or systems to access this information, which can be confusing and hard to manage. To make this easier, we created a new digital platform that puts all the important diabetes data in one place. We designed it together with families of children with T1D and their healthcare providers, so it could truly meet their needs. The goal was to help families and healthcare professionals work together more easily by sharing information, treatment recommendations, and helpful resources. We tested the platform during a six-month pilot study at BC Children's Hospital Diabetes Clinic in British Columbia, Canada. We asked parents and caregivers, as well as healthcare professionals, to share their thoughts through surveys and interviews. In total, 18 caregivers completed surveys, and 11 healthcare providers filled out 41 surveys. At the end of the study, we also interviewed 18 caregivers to hear more about their experience using the platform. The results were promising. Most caregivers (61%) said the platform was helpful in managing their child's diabetes. Over half (56%) felt that their visits with their doctor and diabetes team became more personal and tailored to their child's specific needs. Nearly all of the healthcare providers (90%) said the platform helped improve their clinical visits. Caregivers also shared some deeper insights during interviews. They said the platform helped them feel more prepared for appointments and more confident in managing their child's diabetes. They also noted that the platform worked best when their healthcare team actively used it. Many felt hopeful that this kind of tool could make a big difference in their child's day-to-day diabetes care. In short, this digital platform shows real potential to improve the way families and healthcare teams manage type 1 diabetes together. It could lead to more personalized care, improved communication and connection, and better support for both children and their caregivers. These early results will help us improve the platform and guide how it's used in the future.

## Introduction

Over the past 25 years, managing Type 1 Diabetes (T1D) has undergone a significant transformation driven by technological advancements [1,2]. Young children with T1D are among the quickest to embrace diabetes technologies [3–5]. Increased use of wearable devices, such as Continuous Glucose Monitoring (CGM) systems and insulin pumps enabled the collection of vast amounts of Patient-Generated Health Data (PGHD) that can inform clinical care [6,7]. Accessing these data was challenging for years due to proprietary hardware and software requirements [8]. Recently, device-agnostic platforms created significant opportunities to utilize diabetes data more efficiently [8]. A digital diabetes care ecosystem using PGHD and clinical data can greatly ease the burden on patients, caregivers, and healthcare professional (HCPs) [8]. Furthermore, appropriate access to and use of this data could present unique opportunities to advance research, apply artificial intelligence [9], and enhance quality improvement efforts in healthcare settings.

Digital health apps for T1D provide functionalities like access to manual or automated PGHD (glucose levels, insulin doses), contextual data (food intake, activity), bolus calculators, peer support, and education. Synchronous (remote visits) and asynchronous (chat) HCP communication, along with electronic medical record (EMR) integration, enhance care connectivity [10–17]. Evaluation of apps in pilot and quasi-experimental trials has shown minimal to no impact on glycated hemoglobin (HbA1C) [12,15]. Yet, they improve self-management behaviors like glucose monitoring frequency [12,18,19] and independent patient review of their diabetes data [10,20–22]. When supplemented with remote monitoring and virtual visits, apps have been associated with improved time in range [23], psychosocial health [11], and care experiences [20,24]. Overall, patient and caregiver users have reported high satisfaction with pediatric T1D apps [15,17,18,25] and with the recent rapid growth and adoption of virtual care delivery, patients have come to expect frequent and digitally connected experiences [26,27]. Digital health is, therefore, a rapidly emerging area in T1D [28], but knowledge gaps in implementation and impact are hindering its adoption.

Further research is needed to assess the impact of an interoperable digital ecosystem that ensures secure, seamless health data access for patients, caregivers, and HCPs. Such a system could enhance collaboration and connectivity across clinical visits, improving person- and family-centered care [29]. Integrating PGHD with clinical registries in chronic disease management can reduce administrative burdens, support decision-making, drive research, and improve care quality for more timely, effective treatment [28,30].

### Research goals, aims and objectives

Our team created a platform that integrates various PGHD and clinical data streams. The platform features a front-end digital user interface for patients and their HCP, enabling them to view and collaborate on data while accessing key resources and recommendations. It was developed following a survey conducted with parents and youth (ages 16–17 years) to gain insight into their perspectives on digital trust [31] and to apply user-centered and participatory design approaches during the platform's development, incorporating multiple rounds of iterative user feedback [32–34].

For this pilot study, we launched the 'minimum viable product' (MVP) version of the platform in its first use case for pediatric T1D. There is limited knowledge regarding the real-world evaluation of integrated digital platforms designed to facilitate collaborative management of diabetes. The aim of this pilot study was to gather perspectives from caregivers of children and youth living with T1D and their pediatric HCP on the usability, user experience, and usefulness of the MVP version of this platform.

## Results

### Recruitment

During the two recruitment periods (September 2022 and December 2022), 425 parents of patients at the BCCH Diabetes Clinic received an email invitation to participate in the pilot. In total, 19% (80/425) completed the initial screening

form indicating their interest to participate and 45% (36/80) agreed to receive an emailed consent form, and 29% (23/80) consented to participate (S4 File).

A total of 29 HCP at the BCCH Diabetes Clinic were invited to participate in the pilot study with 83% (24/29) participating in at least one individual or group session; 54% (13/24) consented to participate in the study.

### Retention and survey completion rates

Caregiver participants who had consented and then withdrew (n = 4), did so at the beginning of the study before completing any study activities. Of the 23 caregivers who consented, 83% (19/23) completed all or most of the study activities. While no HCP formally withdrew from the study, work leave and scheduling complexities, including patients' appointment times, impacted how often some HCPs used the platform. As a result, 46% (11/24) of interested and trained HCPs used the platform for their patients' clinic visits and completed post-clinic surveys (S2 File).

Of the 19 participating caregivers, 17 completed the post-onboarding survey, 18 completed the post-clinic survey and end-of-pilot survey, and 15 answered the additional survey questions about app usage. Incomplete responses were treated as missing data, and the denominator was adjusted accordingly in the descriptive analysis.

### Demographic characteristics

Among the 19 caregivers who participated in the study, the median (interquartile range [IQR]) age was 45 [39, 46] years. Most caregivers were white (n = 10/15, 68%) and identified as mothers (14/19, 74%) and were married (11/15, 73%). The median age of their children was 10 [8, 13] years and were living with diabetes for 4.8 (±3.7) years at the time of data collection. Most children (n = 18) used CGM to monitor their blood glucose while one child used a glucometer. Insulin was administered either by insulin pump (n = 10) or through multiple daily injections (n = 9).

### Caregiver perspectives of the platform post-onboarding and use at and between clinical visits

The average System Usability Scale (SUS) score reported by caregivers (n = 17) was 71.9. Table 1 outlines the post-onboarding experience of caregivers (n = 17). Eighteen caregivers shared their insights on using the app before, during, and in between clinical visits. After being onboarded onto the app, 89% (16/18) of caregivers used it to prepare for their visit, and 50% (9/18) reported reviewing their glucose data trends on the app before their clinic appointment. Seventy-two percent (13/18) of caregivers used the clinic visit preparation form (S1 File) and all found it helpful for the clinic visit. Additionally, some caregivers found reviewing and completing tasks (n = 9, 89%), viewing glucose levels (n = 9, 67%) and checking the next appointment (n = 6, 67%) useful for clinic preparation. Overall, 61% (11/18) of caregivers indicated that they found the app helpful or very helpful for the clinic visit, while 28% (5/18) remained neutral, and 11% (2/18) did not find it helpful. Furthermore, 56% (10/18) of participants reported that using the app made their clinic visits and recommendations more personalized.

Most caregivers (94%; 16/17) reported that they would possibly, probably, or definitely use the app to manage their child's diabetes at home following the clinic visit, while 6% (1/17) indicated they would not use it. Caregivers described potential uses for the app at home, including reviewing glucose patterns and adjusting insulin doses, staying connected with their diabetes healthcare team, reviewing resources, and organizing information and tasks related to clinical visits.

When asked how they would like their healthcare team to use the platform between clinic appointments, 61% (11/18) of caregivers indicated they wanted HCPs to review glucose data and provide recommendations for insulin dose adjustments, while 11% (2/18) preferred a brief 15-minute virtual visit with their healthcare team. Additionally, 28% (5/18) expressed a desire for the healthcare team to check in on their or their child's general wellness (e.g., mood) via the platform, and 17% (3/17) were interested in check-ins regarding recent changes in diabetes management (e.g., transitioning from basal/bolus to pump therapy).

After caregivers used the app between clinic visits, they completed a final survey to assess the ease of accessing the platform and viewing data, its security, and its perceived utility for children with T1D (Table 2). Additionally, they reported

**Table 1. Post-Onboarding User Perspectives of Caregivers (n = 17).**

| | Disagree n (%) | Neutral n (%) | Agree n (%) |
|---|---|---|---|
| *Utility* | | | |
| I think it will help me self-manage my child's T1D. | 2 (11.8) | 5 (29.4) | 10 (58.8) |
| *Security* | | | |
| I felt that my child's data was safe and secure. | 0 (0) | 5 (29.4) | 12 (70.6) |
| I appreciated the various security features in place (for example, BC Services card, verified.me, phone password). | 0 (0) | 2 (11.8) | 15 (88.2) |
| *Within App Consent/Assent Process* | | | |
| I found the assent/consent process for connecting devices and sharing diabetes information with physicians informative. | 0 (0) | 6 (35.3) | 11 (64.7) |
| I found the assent/consent process for connecting devices and sharing diabetes information with physicians too long. | 7 (41.2) | 6 (35.3) | 4 (23.5) |
| I found the language in the assent/consent process easy to understand and transparent. | 0 (0) | 3 (17.6) | 14 (82.4) |
| I understood why I was consenting (and my child was assenting, if applicable). | 0 (0) | 1 (5.9) | 16 (94.1) |
| I found the child assent process helpful in making my child proactive about their health.* | 0 (0) | 9 (56.3) | 7 (43.8) |
| I found it helpful to be guided in a discussion through the app about consent/assent and exactly where my child's diabetes data goes. | 0 (0) | 4 (23.5) | 13 (76.5) |
| My child and I enjoy controlling who can see diabetes data/devices and that I can change this at any time. | 0 (0) | 7 (41.2) | 10 (58.8) |

BC, British Columbia; T1D, type 1 diabetes.

* n=16, question left blank by one respondent.

how frequently they used the Careteam platform, with 67% (10/15) using it at least once every 1–2 months and 20% (3/15) at least once per week, while none reported daily use.

### Healthcare providers

A total of 11 HCP agreed to participate in the pilot study and completed post-clinic surveys to gather their perceptions on the usefulness of the web-based application before and during clinical visits with study participants. At baseline, HCP predominantly had experience using Dexcom Clarity and Glooko to visualize CGM and pump data. A total of 41 post-clinic surveys were completed by 11 participating HCP, averaging 3–4 surveys per HCP. Overall, 90% (37/41) of survey responses indicated that the HCPs were satisfied with using the HCP-user interface to support clinical visits. When asked if the application improved pre-visit preparation and enhanced patient interactions during clinic visits, 90% (37/41) and 88% (36/41) of respondents, respectively, answered "yes." The application features deemed most valuable by HCPs included the clinical visit preparation form, tasks and reminders (e.g., uploading insulin pump information), and access to glucose and insulin dose data (Fig 1).

When asked how they might use the application in the future to connect with patients between visits, 71% (29/41) indicated they would send a check-in regarding a recent change in diabetes management (e.g., transitioning from basal/bolus to pump therapy), 41% (17/41) would send a check in on general wellness, and 46% (19/41) would review glucose data and provide recommendations on insulin dose adjustments.

### Caregiver perspectives on the app

Qualitative interviews (n = 18) conducted at the conclusion of the study revealed three key themes that highlighted caregivers' experiences with the platform: 'convenient connection', 'usage drives value', and 'hope and future benefit'. Each

**Table 2. Caregivers' (n = 18) Perceptions of the Ease of Access, Security, and Utility of the platform.**

| | Disagree n (%) | Undecided n (%) | Agree n (%) |
|---|---|---|---|
| *Ease of Access* | | | |
| Logging in to my app account is easy. | 1 (5.6) | 2 (11.1) | 15 (83.3) |
| Viewing my/my child's clinical information and diabetes data in the Careteam app is easy. | 2 (11.1) | 2 (11.1) | 14 (77.8) |
| *Security* | | | |
| I feel that my/my child's personally identifying information (i.e., name, birthdate) and clinical data (i.e., blood glucose) is secure in this app. | 0 (0) | 6 (33.3) | 12 (66.7) |
| *Utility* | | | |
| I would recommend use of this app to other families with children living with T1D. | 0 (0) | 5 (27.8) | 13 (72.2) |

T1D, type 1 diabetes.

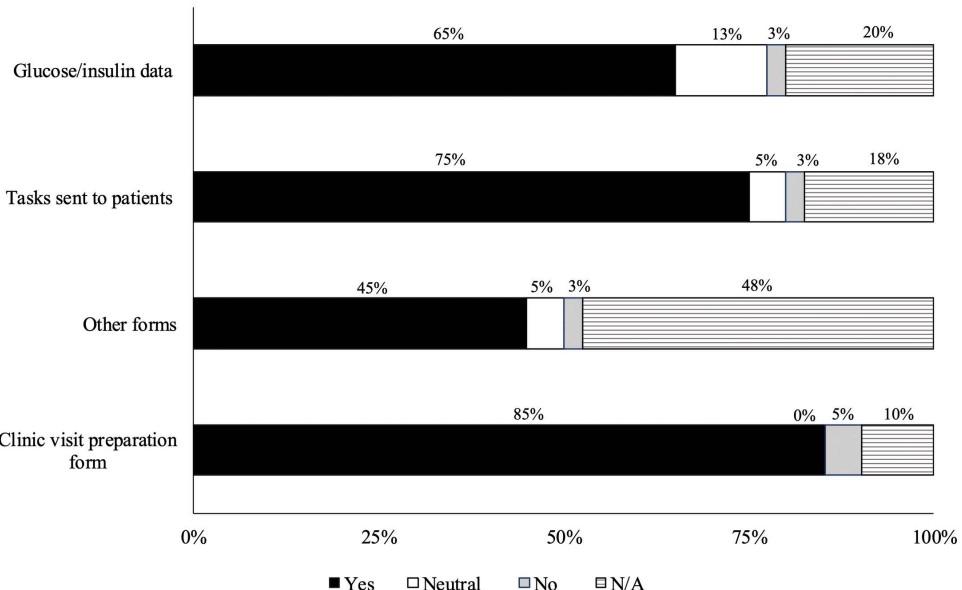

**Fig 1. Usefulness of the Features Available in the Web-Based Application for Healthcare Professionals.**

theme is supported by sub-themes that provide a deeper understanding of caregivers' experiences. The outcomes of the qualitative interviews, including themes, sub-themes, and quotations, are detailed in Fig 2 and S5 File.

**Theme 1: Convenient connection**

The first theme that emerged from the qualitative interviews was 'convenient connection'. Caregivers described the platform as convenient and handy (subtheme 1) because it connected them to their child's diabetes device data, other diabetes-related resources, and the diabetes healthcare team all in one place. They valued the convenience of having their child's data and important information in a portable, easily accessible format, ensuring access anywhere, while appreciating that a single, accessible source for the healthcare team eliminated worries about forgetting papers and information at home. As one caregiver noted, "*It has the potential to be a really useful on hand tool with readily available*

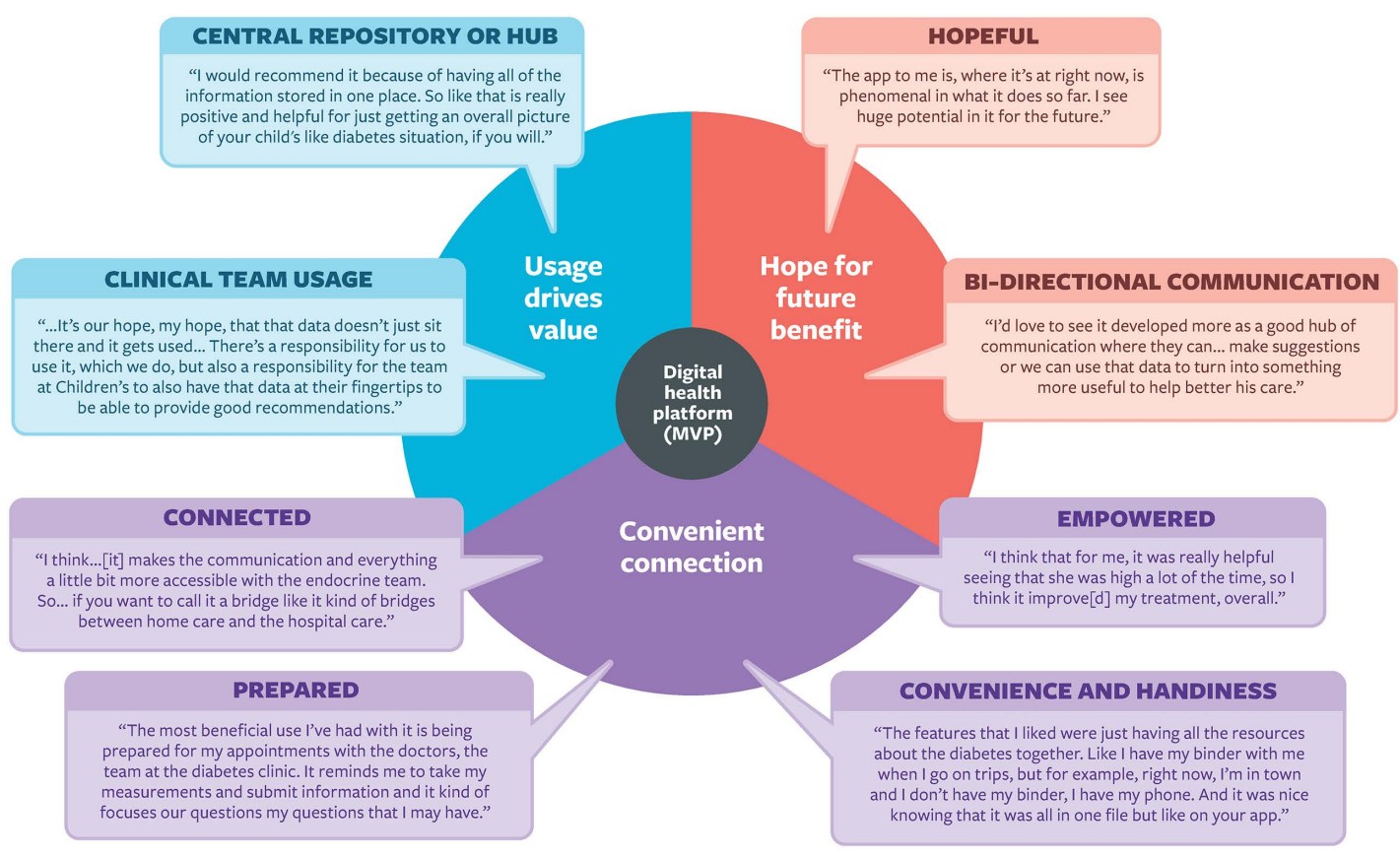

**Fig 2. Themes, Subthemes and Quotes from the Qualitative Interviews with Caregivers.** MVP, minimum viable product.

*information. And it's easy if you have 5 minutes just to quickly look at perhaps [child's], you know, the data or just the prompts...rather than everything getting lost in the hundreds of junk emails that you get every day."*

Caregivers also reported feeling more connected (subtheme 2) to the healthcare team through the platform. Patients and caregivers usually receive tasks, forms, and reminders by mail or email, while HCP recommendations provided during visits often go undocumented, having to rely on memory or notes. With the platform, they received forms and reminders before appointments, and HCP recommendations were recorded afterward, strengthening their connection with the care team. Another quote captured this sentiment: *"It was beneficial, for sure because it was...personal... It was a message that [my child's] doctor actually entered to follow up after the appointment and touch base... it was a friendly note with the recommendations...a little bit of a better connection because it was like he took the time to...send us the information. Instead of, at the previous appointments, I would have to make sure to write it down and remember it."*

Caregivers felt empowered (subtheme 3) to manage their child's T1D using the platform's glucose trends, allowing them to review data, adjust insulin doses, and make care decisions without waiting for an HCP appointment. They credited the platform for supporting self-management by providing everything in one place. One caregiver shared, *"When we look at the 30 day and the 90 day...I like to be able to see that and...your glucose levels and your time in range and all that type of thing that's neat to see...we don't look at this data in any way, shape or form as a sort of report card...when we analyze the graph and look at them and go... 'in the morning, you're doing this, maybe we should tweak that.' That truly is*

*very helpful and helps us understand."* The platform was also seen as a potential opportunity for caregivers to involve their child in diabetes self-management and gain the skills and confidence necessary for independent self-management.

Nearly all caregivers felt more prepared (subtheme 4) for clinic appointments, appreciating the ease of completing forms electronically without searching emails or remembering to bring paperwork. They also indicated that this convenience translated into the healthcare team being better prepared. As noted by one caregiver, *"In the past when I filled in the paper and brought it to the appointment it was… probably not enough time for the nurses or dietitians, whoever is going to see us that time...for them to grab that paper and review it. Once, you know, we come to the office and then our appointment is right away, so it's better if they can see it on the app and they can review it before we come."* Additionally, completing the form prompted caregivers to consider their questions for the healthcare team, further reinforcing their sense of preparedness for clinic appointments. One caregiver said, *"I used it in preparation for the appointment...So it kind of helped me prepare a little bit better for the appointment, because I guess it made me have to sit and think for a second on like questions that I had for my clinic appointment. So those prompts before the appointment were actually helpful and I used it."*

**Theme 2: Usage drives value.**  This theme highlighted that, caregivers found the platform most beneficial when the healthcare team actively engaged with it (subtheme 1). Receiving tasks and recommendations from HCPs was valuable, and many wanted more prompts, support, and personalized information. They stressed the importance of both caregivers and HCPs actively using the data rather than letting it sit idle. As one caregiver noted, *"I think what would be more helpful to me is if on the... healthcare team side that they were pushing the information to me more...here's your next appointment confirm it through the app. Here's some reminders that you need to get this blood work done by this time...I would love that information to be pushed to me more instead of me inputting it...because that would be more helpful...Reminders on the blood work...Celiac...She had her A1C done at our last appointment and I have a paper copy of that but I don't have that anywhere in my Careteam or attached to our last visit."*

Caregivers saw the platform as a central repository or hub for data (subtheme 2), emphasizing its value in sharing key information—personal details, educational resources, and diabetes-related apps—with the child's circle of care. They viewed having all members access the same data in one place as a significant advantage. One caregiver summarized this by saying, *"For us, it's not my daughter who's taking care of her diabetes, it is really a team. It's myself, my husband, my parents, her endo team, and we can all use that as kind of a you know a hub."*

**Theme 3: Hope for future benefit.**  The final theme identified was 'Hope for Future Benefit'. Most caregivers wanted to communicate with the healthcare team through the platform to have their questions answered between clinic appointments. While email and phone calls are the current methods of communication for patients and caregivers with the clinic, many participants considered these methods to be formal, which made the healthcare team feel inaccessible. For example, a message centre supporting bidirectional communication (subtheme 1) would further centralize information in one place, providing a system to enhance collaboration. This was highlighted by a caregiver, who stated, *"…So it would be, to have a two-way street in terms of the care that we're giving to our son and that they can give to him, in terms of like, hey, I've noticed this over the last three weeks this has been happening, here's what you should do to bring those numbers down, or bring them up, or whatever. So I'd love to see it developed more as a good hub of communication where they can action, where they can make suggestions, or we can use that data to turn into something more useful to help better his care."*

Participants expressed a greater likelihood of asking the team if they could send a direct message through the platform and receive a reply asynchronously. The ability to pose a question when it arose was considered valuable, with caregivers highlighting that it could enhance their self-management by obtaining knowledge from an HCP. Furthermore, caregivers desired bidirectional communication, where the healthcare team would remotely monitor their child's data and periodically reach out between appointments to offer suggestions, advice, or check in with the family. As mentioned by another caregiver, *"I think it could be utilized more. I think that they could be, you know communicating, especially with some of*

*the little ones where, you know, we have to make frequent changes more with their insulin doses. To communicate, you know, instead of like every 3 months maybe to talk to like somebody via the app, a nurse or something, to say, hey, you know, maybe make a change here or a change there. I feel like that could be more beneficial. I mean, these little kids they tend to need changes so frequently as they are growing and changing, and as a parent, I feel like I'm still scared to make those changes without kind of the guidance and reassurance from a nurse."* Many caregivers felt hopeful (subtheme 2) that the platform would enhance the care and self-management of their child with T1D, seeing potential for the platform in the future. They emphasized the importance of integrating the platform with existing systems (i.e., other patient-facing apps for caregivers and EMRs for HCP). The platform patients/caregivers used during the pilot was regarded as merely the starting point; further development and enhancements would make the platform even more valuable.

## Discussion

### Summary

We developed and piloted an MVP of a digital health platform where patients, caregivers, and pediatric diabetes HCP could collaborate on T1D diabetes management. Most caregivers and HCP users on our platform reported that it was useful and supported a more personalized care experience. Caregivers described the platform as convenient, stating that it fostered a better connection with their healthcare team and made them feel more prepared for clinic visits while empowering them to self-manage their child's diabetes. The platform features of being able to review glucose data trends and preparation for clinic visits were seen as most helpful by caregivers and HCP. Evaluation of system usability of our platform indicated it had 'good usability,' and scored above the published benchmark score [35]. Our collaboration with end users throughout the platform development process [31,33] likely contributed to users reporting high usefulness and acceptable usability, as this co-creation approach has been recognized as essential for successful digital health technology development [36,37]. Our study highlights the potential of digital technology in enhancing the delivery of person- and family-centred care [29].

### Interpretation and comparison with existing literature

Enhancing patient preparedness through pre-visit planning has been associated with improved relationships between patients and HCPs, increased patient empowerment, and active participation in chronic disease self-management [38], all essential for achieving person-centred care [38]. For both patients/caregivers and HCPs in our study, the most helpful features of the platform were the clinic visit preparation form, tasks (e.g., getting blood work done), and reminders (e.g., uploading insulin pump data and entering MDI doses), which were sent prior to a clinic visit. A systematic review found that over 80% of studies showed pre-visit planning improves patient-provider communication, patient knowledge, illness perception, and care satisfaction. It also saves time during visits, enabling focus on more critical issues [39].

Caregivers in our study reported that using the platform strengthened their connection with the clinical care team. Strong patient-provider relationships have long been acknowledged as crucial for delivering high-quality, person-centred care [40–43]. Our qualitative data show that HCP engagement is key for patients to benefit fully from the platform, and bidirectional communication enhances its value. Studies on remote patient monitoring confirm the importance of HCP involvement with digital tools [44–48].

Numerous studies have highlighted the benefits of remote patient monitoring between clinic visits in detecting health deterioration earlier and enabling prompt intervention, particularly for individuals with chronic conditions [44–50]. In one study, Glooko [51], a commercially available population health platform, generated monthly reports identifying participants with frequent hypo- and hyperglycemia, who then received personalized self-management advice from HCPs via phone or secure EMR messaging. Most found the intervention helpful, preferring EMR messages over phone calls and favoring insulin dose guidance over other support. The study also reported a -0.25% difference in the glucose measurement index, a CGM-based proxy for glucose control, between the remote monitoring and control group [52].

In another study at a US pediatric diabetes clinic, a team developed and implemented a patient prioritization algorithm using CGM data from patients with newly diagnosed T1D [53,54]. A subset of patients in the intervention received remote monitoring through EMR integration, where a diabetes educator prioritized their review and provided secure self-management recommendations. Compared to a historical cohort of newly diagnosed T1D patients not using CGM, those using CGM had lower HbA1C levels at 6-, 9-, and 12-months post-diagnosis. Among CGM users, those who also received HCP-guided remote monitoring had even lower HbA1C levels, highlighting the added benefit of HCP involvement in digital tool use [53].

In a 6-month study on remote care for children living with T1D [55], on-demand clinician visits combined with the TidePool data visualization platform increased references to device data and improved patient engagement. Providers valued its integrated data, user-friendly interface, remote access, and educational benefits but recommended automating uploads and EMR integration for further enhancement [55].

Our study participants valued bi-directional communication via messaging for remote patient monitoring; while this can improve patient quality of life [56], literature gaps limit understanding of implementation barriers [57] and patient outcomes. Our pilot study demonstrated the potential of integrating digital health with in-person diabetes care, fostering greater connection between patients, caregivers, and HCPs for a more person-centered model. Scaling clinical implementation will require change management strategies [58], service design methods [59] and integration with existing systems like EMRs to optimize HCP adoption.

## Strengths and limitations

The main strength of this pilot study is the development of the MVP and its successful deployment in a real-world pediatric diabetes clinic. Caregivers and HCPs were highly engaged, contributing robust survey responses and qualitative insights that strengthened our data. Our mixed-methods approach provided a deeper understanding of successes, challenges, and areas for improvement, offering insights valuable not only for our platform but also for other digital health tools supporting childhood diabetes and chronic disease care. These qualitative findings will inform future platform enhancements, including new features and functionalities.

Our study is limited by its small, homogeneous sample, which introduces a biased perspective and limits generalizability to the broader population of patients living with T1D. We collected only caregiver feedback (not patient feedback) on the platform, which restricts understanding from a child's or youth's viewpoint. In this small pilot study, we could not focus adequately or gather enough data to ensure an equitable digital diabetes ecosystem [60], as our participants did not represent the full range of contexts and circumstances experienced by people living with diabetes. Another limitation of our pilot study is the lack of detailed technical usability metrics, including login difficulties, onboarding friction, and connectivity or sync issues.

## Future directions

Our next steps include evaluating an improved version of the platform in a type 1 hybrid effectiveness implementation quasi-experimental trial at BC Children's Hospital. During this, we will expand recruitment so that our sample reflects the ethnic distribution of T1D in Canada through mixed recruitment methods with the goal of reaching diverse socioeconomic groups. We will also assess digital literacy to explore adoption and barriers across literacy levels. We plan to enroll youth participants aged 15–17 who are primary decision-makers in their diabetes care alongside a broader cohort of younger children whose parents remain the primary managers.

## Conclusion

Through this research, we have moved closer to developing a digital ecosystem where technology provides seamless and secure access to clinical data and PGHD for patients, caregivers, and HCP, while also offering a secure digital pathway for patients and HCP to stay connected and collaborate on care before, during, and between clinical encounters. Future

research is necessary to determine if our platform can enable equitable, efficient, and effective care coordination while supporting data-informed decision-making, ultimately improving patient outcomes, reducing administrative burdens, and advancing research and quality improvement.

## Materials and methods

### Development

Between 2020 and 2022, our team, consisting of a public-private partnership, developed a platform that allowed patients to securely connect their personal wearable devices such as insulin pumps and CGM, view near real-time health data through continuous synchronization, and access Careteam Technologies [61] (referred to as 'Careteam' herein), a front-end digital health app for healthcare coordination that was tailored for managing pediatric T1D. The design and development of the platform were guided by extensive formative research [31,33], including multiple rounds of usability testing (manuscripts in preparation) with end-users (i.e., patients and caregivers living with T1D and their HCP). The development included an elaborate back-end architecture that utilized standard application programming interfaces and transformed data from various sources using fast healthcare interoperability resources [28,62,63].

### Ethics approval

After receiving a waiver from the Children & Women's Research Ethics Board (REB) at the University of British Columbia, this study was approved under the Quality Improvement category by the Data Collections and Solutions Steering Committee and the Provincial Health Services Authority Privacy Office through the completion of a Privacy Impact Assessment. Informed consent was obtained from each parent/guardian and HCP for their participation. Additionally, assent was obtained from minors (aged over 7 years). This paper adheres to the Standards for Quality Improvement Reporting Excellence (SQUIRE) 2.0 guidelines for reporting quality improvement studies [64].

### Study design and context

We conducted a six-month observational pilot study at the Diabetes Clinic at BC Children's Hospital (BCCH) in Vancouver, Canada, to assess the usefulness of implementing the digital platform in a real-world clinical care context. As the province's only tertiary pediatric hospital, BCCH serves 850 of the 2,300 children with T1D [65], while the rest receive care at community-based clinics supported by local pediatricians and endocrinologists.

### Intervention

In this pilot study, the intervention included a digital app for patients and caregivers, and a web-based Careteam application for HCP. Caregivers used the app to view their child's clinical data, while HCPs accessed the Careteam application via single sign-on with hospital credentials to manage diabetes care plans and dashboards.

The account set-up and onboarding process for caregivers was facilitated by a research coordinator via Zoom (Zoom Communications Inc, San Jose, CA) who offered support for app installation, set-up, identity authentication and an additional consent/assent process to use the app in clinical care. Users could then securely link their diabetes devices, such as an insulin pump, glucose sensor, and glucometer, enabling continuous data synchronization and near real-time visualization of glucose trends on the Careteam interface (Fig 3A). The research coordinator created HCP accounts and provided training at the study's launch. Through Careteam, caregivers accessed educational resources, viewed HCP-assigned tasks (e.g., complete blood work), and completed pre-visit surveys (e.g., clinic preparation form, S1 File). They could visualize HCP recommendations for insulin adjustments, nutrition, and lifestyle or behaviour changes. Patients on multiple daily injections could manually enter basal insulin doses, insulin-to-carb ratios, and sensitivity factors, with each entry timestamped by user, date, and time (Fig 3B). Through the web-based application, HCPs could review continuous

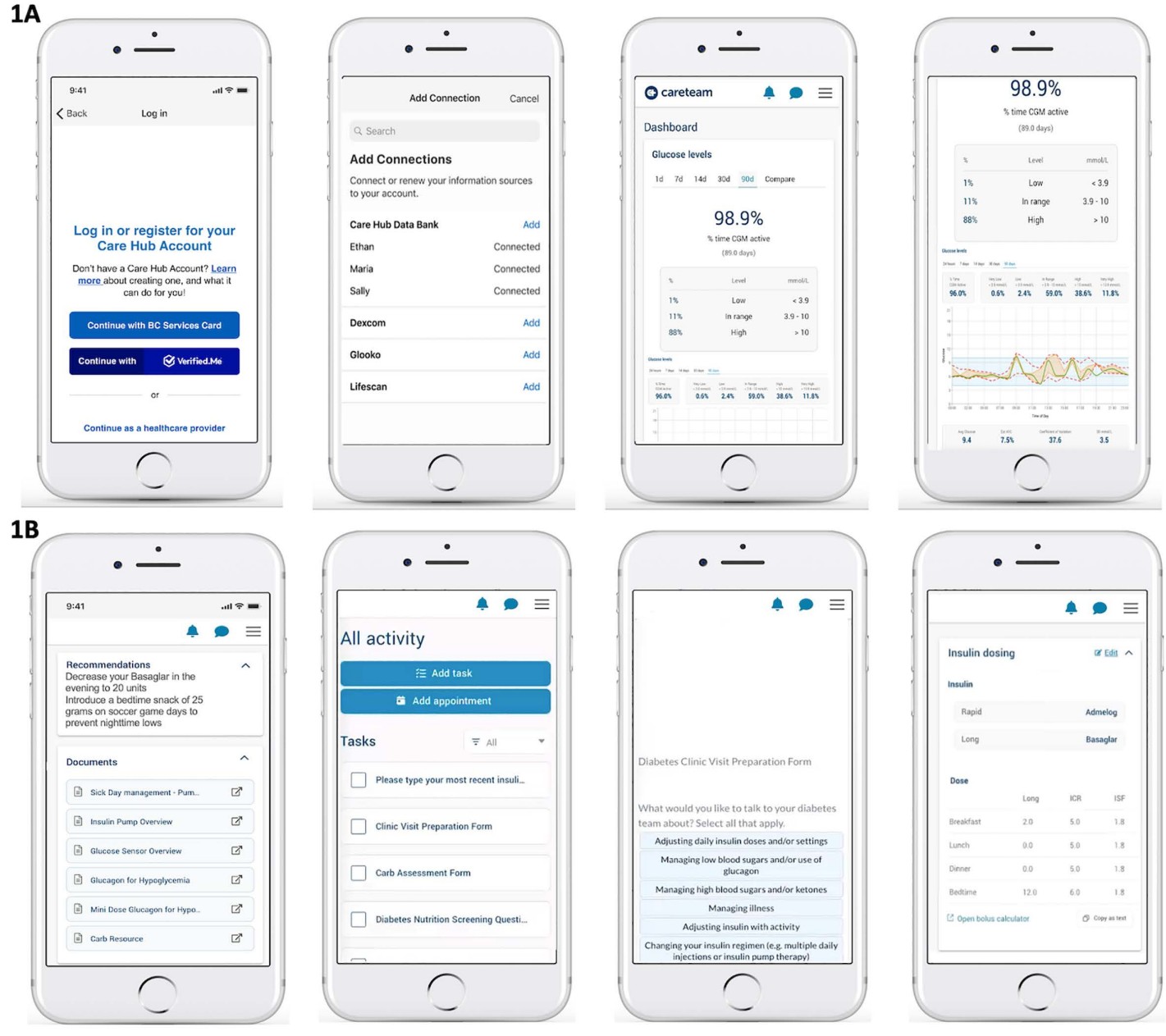

**Fig 3. Screenshots From the Patient-Facing App and Careteam's Shared Patient Care Plans and Dashboard: A: Onboarding and Glucose Monitoring Data Visualization; B: HCP Recommendations, Tasks, Diabetes Clinic Visit Preparation Form, and Multiple Daily Injection Doses.**

glucose data trends, access pump settings or MDI doses, input recommendations, and assign tasks (Fig 4). HCPs and patient/caregiver participants were instructed to use the web-based application or smartphone app before, during, and between regularly scheduled clinical visits throughout the 6-month pilot study (as detailed in Fig 5). The platform was monitored to address minor technical issues, and user feedback informed instructional text improvements for a better experience.

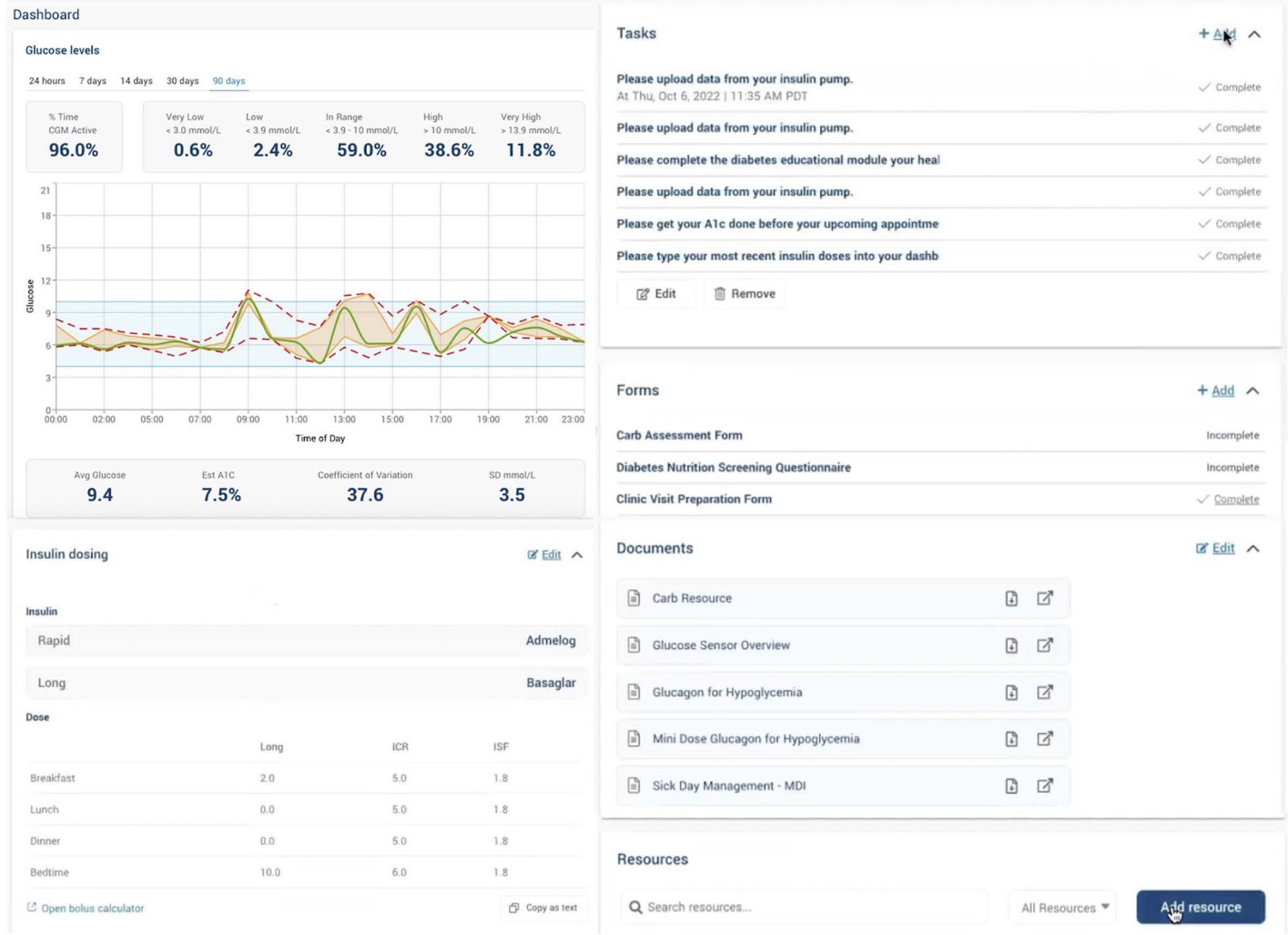

**Fig 4. Web-Based Application for HCP.**

## Recruitment

BCCH Diabetes Clinic HCPs including pediatric endocrinologists, nurses, and dietitians were contacted by a research coordinator who explained the study and offered group and individual training sessions before obtaining their consent to participate. Potentially eligible patients and their caregivers were identified and contacted via email using a clinical database maintained by the Division of Endocrinology's Diabetes Clinic and were recruited throughout the pilot study; we aimed to onboard patient and caregiver participants 1–3 weeks before their follow-up appointment at the clinic. Eligibility criteria is presented in Fig 6.

Caregivers completed a secure online screening form to express their interest in participating in the study and to confirm their eligibility. Once eligibility was confirmed, consent to participate in the pilot study was obtained from parents (on behalf of their children) through an emailed electronic Portable Document Format (PDF) consent form using REDCap. Consent from HCP was similarly obtained via electronic PDF using REDCap.

| | Review Glucose Trends | | Pre-appointment Tasks | | Update Insulin Doses or Pump Settings | | Pre-appointment Questionnaires | | Healthcare Provider Recommendations | |
|---|---|---|---|---|---|---|---|---|---|---|
| | Patient/ Caregiver | Healthcare Provider | Patient/ Caregiver | Healthcare Provider | Patient/ Caregiver | Healthcare Provider | Patient/ Caregiver | Healthcare Provider | Patient/ Caregiver | Healthcare Provider |
| **Before Clinic Appointment** | ✔ | ✔ | ✔ | ✔ | ✔ | ✔ | ✔ | | | ✔ |
| **During Clinic Appointment** | ✔ | ✔ | | | ✔ | ✔ | | | | ✔ |
| **Between Clinic Appointment** | ✔ | ✔ | | | ✔ | ✔ | | | | ✔ |

**Fig 5. Possible Actions by HCP and Caregivers Using the Dashboard/App Before, During, and After Clinic Appointments.** HCP, healthcare professionals. Tasks included caregivers entering current MDI doses, uploading pump, or completing a pre-appointment questionnaire; recommendations entered into the dashboard by HCP including changes to insulin dosing regimens, diet/activity, etc.

## INCLUSION

✔ at least 3 years old and no older than 18 years old at the time of recruitment
✔ followed by a BC Children's Hospital pediatric endocrinologist who consented to participate in the study
✔ not transitioning to adult care during the study period
✔ at least six months since their T1D diagnosis
✔ had at least one clinical appointment scheduled during study period
✔ using or willing to use (provided free of charge to ensure equity) Lifescan Verio Reflect glucometer, Dexcom G6 CGM, and the Omnipod insulin pump
✔ had a substitute decision-maker or caregiver who was able to understand and communicate effectively in both written and spoken English

## EXCLUSION

✖ using a non-Health Canada approved open-source automated insulin delivery system ('Do-It-Yourself looping')
✖ diagnosed with multiple or complex conditions, or if receiving active treatment for a significant mental health diagnosis, such as an eating disorder or depression

**Fig 6. Study Eligibility Criteria.**

## Data collection

The research coordinator monitored participant recruitment and retention. Feasibility was assessed by counting HCPs who attended training sessions and caregivers who responded to the eligibility screening form.

Data were collected from caregivers and HCPs at multiple time points. At baseline, an anonymous demographic survey gathered patient and caregiver information. After a facilitated onboarding session, caregivers completed a post-onboarding survey, including the System Usability Scale (SUS) [66–69] to assess usability. Additional questions captured initial impressions of the app's utility, security, and onboarding experience, with responses on a 5-point Likert scale.

Following their first clinic appointment using the app, caregivers completed a researcher-developed survey (S2 File) on app usage and clinical utility. HCPs also completed a survey after each clinic to gather their perspectives on the platform's use and utility for clinical care. After their final clinic appointment during the study period, participants provided their overall perspectives and experiences using the app through a researcher-developed survey (S3 File). On average, time from the last clinic visit to survey completion was 6.8 weeks. At the end of the study, semi-structured qualitative interviews were conducted to gather in-depth reflections on the app's usability, user experience, and its usefulness in managing diabetes.

Surveys for caregivers and HCP were administered and completed online using REDCap [70,71]. Caregivers accessed surveys via an emailed link, while HCPs completed them on a tablet provided by the research team in the clinic. Researcher-developed survey questions were reviewed by a cross-disciplinary team but not formally evaluated. Caregivers received a CAD 25 gift card per survey (except at baseline) and CAD 50 for interviews, while HCPs received CAD 5 per survey.

## Data analysis

Descriptive quantitative data analysis was conducted using STATA 15.1 (StataCorp, College Station, TX). Data were presented as medians, and proportions were calculated to characterize the population's demographics. Responses to surveys were analyzed using descriptive statistics. Zoom audio recordings of qualitative interviews were used to generate accurate verbatim transcripts of the interviews. Data gathered from qualitative interviews were analyzed using NVivo (R1.7; Lumivero, Denver, CO), where ST, SP, and CZ inductively coded the transcripts using Braun & Clarke's thematic analysis methodology [72]. SP, CZ, and ST independently coded four transcripts before meeting to develop a coding scheme. They then tested the scheme on one additional transcript each and reconvened to refine codes. The finalized coding scheme was used to systematically code all transcripts, identifying themes and sub-themes. Any coding disagreements were resolved through discussions. Interview transcripts were also pragmatically analyzed [34] to inform continued app development and improvements, including suggestions for additional features and any usability issues that arose during app onboarding or use.

## Supporting information

**S1 File. Clinic Visit Preparation Form.**
(DOCX)

**S2 File. Post Clinic Appointment Survey.**
(DOCX)

**S3 File. Caregiver End of Phase Questionnaires.**
(DOCX)

**S4 File. Participant Flow.**
(DOCX)

**S5 File. Qualitative Interview Themes, Subthemes and Quotations.**
(DOCX)

## Acknowledgments

We thank all the individuals with lived experience (caregivers, children, and youth) and HCPs at BC Children's Hospital who participated in this study. The authors extend their appreciation to the University of British Columbia, BC Children's Hospital Research Institute, BC Children's Hospital Foundation, and Canada's Digital Technology Supercluster for their support and contributions to the success of this project. We are grateful to our industry partners, including Careteam Technologies, IDENTOS, and Smile Digital Health, whose contributions (intellectual, financial, and in-kind) were instrumental in developing the platform. We also acknowledge members of the TrustSphere Collaborative: Shazhan Amed, Wyeth Wasserman, Tibor van Rooij, Matthias Görges, Bonnie Barrett, Susan Pinkney, Elizabeth M Borycki, Andre Kushniruk, Holly Longstaff, and Alice Virani.

## Author contributions

**Conceptualization:** Shazhan Amed, Anila Virani, Matthias Görges, Bonnie Barrett, Tibor van Rooij, Elizabeth M Borycki, Andre Kushniruk, Holly Longstaff, Alice Virani, Wyeth W Wasserman.

**Formal analysis:** Susan Pinkney, Carlie Zachariuk, Sukhpreet K Tamana, Elizabeth M Borycki, Andre Kushniruk.

**Funding acquisition:** Shazhan Amed, Bonnie Barrett.

**Investigation:** Shazhan Amed, Susan Pinkney, Fatema S Abdulhussein, Carlie Zachariuk, Sukhpreet K Tamana, Elizabeth M Borycki, Andre Kushniruk.

**Methodology:** Shazhan Amed, Susan Pinkney, Fatema S Abdulhussein, Elizabeth M Borycki, Andre Kushniruk.

**Project administration:** Bonnie Barrett.

**Supervision:** Shazhan Amed, Anila Virani.

**Visualization:** Shazhan Amed, Susan Pinkney, Carlie Zachariuk.

**Writing – original draft:** Shazhan Amed.

**Writing – review & editing:** Susan Pinkney, Fatema S Abdulhussein, Anila Virani, Carlie Zachariuk, Sukhpreet K Tamana, Shruti Muralidharan, Matthias Görges, Bonnie Barrett, Tibor van Rooij, Elizabeth M Borycki, Andre Kushniruk, Holly Longstaff, Alice Virani, Wyeth W Wasserman.

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
