## [Decision Letter · Decision Letter 0]

29 May 2025

Response to Reviewers
Revised Manuscript with Track Changes
Manuscript
**Journal Requirements:**
**Additional Editor Comments (if provided):**

**Reviewers' Comments:**

**Comments to the Author**

1. Does this manuscript meet PLOS Digital Health’s publication criteria?

Reviewer #1: Yes

2. Has the statistical analysis been performed appropriately and rigorously?

Reviewer #1: Yes

3. Have the authors made all data underlying the findings in their manuscript fully available (please refer to the Data Availability Statement at the start of the manuscript PDF file)?

Reviewer #1: Yes

4. Is the manuscript presented in an intelligible fashion and written in standard English?

Reviewer #1: Yes

Reviewer #1: I read with great interest the manuscript "Caregiver experiences of an integrative patient-centered digital health application for pediatric type 1 diabetes care: findings from a pilot clinical trial" by Amed et al. I truly enjoy the ambitious approach to both developing and carefully testing a joint agnostic platform for diabetes data which is truly a need in clinical diabetes management. After reading the manuscript I only have a few minor questions that I think could be worth adressing:

1. Could you please provide descirptive data of the patients included, i.e. age, age at diagnosis etc and also previous use of CGM and/or insulin pump

2. Could you please provide descriptive data on the HCPs and their previous experience from working with CGM data and other platforms.

3. Did the HCPs or caregivers have previous experience from other platforms such as Tidepool and in what way does your MVP platform differ from such platforms.

**Do you want your identity to be public for this peer review?** For information about this choice, including consent withdrawal, please see our Privacy Policy

Reviewer #1: **Yes: ** Daniel Espes, MD. PhD

**Figure resubmission:****Reproducibility:** To enhance the reproducibility of your results, we recommend that authors of applicable studies deposit laboratory protocols in protocols.io, where a protocol can be assigned its own identifier (DOI) such that it can be cited independently in the future. Additionally, PLOS ONE offers an option to publish peer-reviewed clinical study protocols. Read more information on sharing protocols at https://plos.org/protocols?utm_medium=editorial-email&utm_source=authorletters&utm_campaign=protocols

---

## [Decision Letter · Decision Letter 1]

9 Jul 2025

Response to Reviewers
Revised Manuscript with Track Changes
Manuscript
**Journal Requirements:**
**Additional Editor Comments (if provided):**
**Reviewers' Comments:**

**Comments to the Author**

Reviewer #1: All comments have been addressed

Reviewer #2: All comments have been addressed

publication criteria?

Reviewer #1: Yes

Reviewer #2: Yes

3. Has the statistical analysis been performed appropriately and rigorously?

Reviewer #1: Yes

Reviewer #2: Yes

4. Have the authors made all data underlying the findings in their manuscript fully available (please refer to the Data Availability Statement at the start of the manuscript PDF file)?

Reviewer #1: Yes

Reviewer #2: Yes

5. Is the manuscript presented in an intelligible fashion and written in standard English?

Reviewer #1: Yes

Reviewer #2: Yes

Reviewer #1: (No Response)

Reviewer #2: Thank you for the opportunity to review this insightful manuscript. I commend your team for developing, deploying, and evaluating this patient-centered digital health platform for pediatric T1D population. This real-world pilot study is methodologically sound, and your mixed-methods approach (blending quantitative survey data with rich qualitative insights) offers a compelling view into caregiver experiences, perceived value, and implementation challenges. It is evident that this work is the result of a thoughtful, collaborative, and multidisciplinary effort.

Your manuscript is well-written, engaging, and clearly addresses an important gap in digital health for chronic disease management, particularly in pediatrics, where caregiver engagement is vital. The emphasis on co-design with end users, iterative feedback integration, and usability testing reflects strong alignment with human-centered design principles. Caregivers' voices come through clearly in your qualitative findings, and the themes you've presented (convenient connection, usage drives value, and hope for future benefit) are both meaningful and actionable.

That said, I'd like to offer several suggestions that may help strengthen the paper further and increase its utility for readers interested in implementation science and digital health innovation:

1. While the manuscript acknowledges the sample's demographic limitations, a more detailed reflection on how future phases might ensure greater diversity (ethnic, socioeconomic, technological literacy) would be valuable. This is especially important in digital health, where equity and access are persistent challenges.

2. The focus on caregivers is understandable for a pediatric population, but a discussion on why children or adolescents weren't interviewed, and whether they will be involved in future evaluations, would enrich the perspective.

3. Including even brief commentary on technical usability like login issues, sync failures, or onboarding difficulties, would offer readers a fuller picture of real-world feasibility.

4. Your paper would benefit from a clearer breakdown of how different healthcare provider roles (e.g., endocrinologists vs. nurses vs. dietitians) engaged with the platform and how their workflows may have been affected.

5. Given that most caregivers reported using the app monthly or less, it would be helpful to reflect on how the platform might maintain engagement over time, especially after the novelty of a pilot wears off.

6. While the study is practical and applied, there is little engagement with behavioral health or technology adoption theory (e.g., Technology Acceptance Model, Health Belief Model, COM-B). Consider suggesting a theoretical frame to contextualize caregiver behaviors and attitudes, which may help guide future iterations or hypothesis-driven trials.

7. The architecture's mention of (Fast Healthcare Interoperability Resources) suggests alignment with interoperability standards, which is appreciated. However, only minimal technical detail is provided. Consider expanding slightly on how FHIR was implemented in this context, for example, whether it enabled integration with EMRs, how data security or identity management was handled, and whether institutional or technical challenges affected scaling the platform beyond BCCH.

8. The feedback on the consent/assent process (e.g., length, transparency) is valuable. It would be great to see how your team plans to iterate on this process, especially as you consider broader rollout.

Despite these suggestions, I found the manuscript to be a thoughtful and meaningful contribution. It's encouraging to see platforms designed with and for caregivers and healthcare teams that aim to improve communication, data visibility, and shared decision-making. Your findings not only provide a foundation for further platform development, but also offer useful insights to the broader digital health community working toward person- and family-centered care.

Thank you again for the opportunity to review this important work. I hope my comments are helpful as you finalize your manuscript, and I look forward to seeing the platform's evolution and continued impact in future studies.

**Do you want your identity to be public for this peer review?** For information about this choice, including consent withdrawal, please see our Privacy Policy

Reviewer #1: **Yes: ** Daniel Espes

Reviewer #2: **Yes: ** Nour Kassem

**Figure resubmission:****Reproducibility:** To enhance the reproducibility of your results, we recommend that authors of applicable studies deposit laboratory protocols in protocols.io, where a protocol can be assigned its own identifier (DOI) such that it can be cited independently in the future. Additionally, PLOS ONE offers an option to publish peer-reviewed clinical study protocols. Read more information on sharing protocols at https://plos.org/protocols?utm_medium=editorial-email&utm_source=authorletters&utm_campaign=protocols

---

## [Decision Letter · Decision Letter 2]

27 Aug 2025

Caregiver experiences of an integrative patient-centered digital health application for pediatric type 1 diabetes care: findings from a pilot clinical trial

PDIG-D-25-00235R2

Dear Dr. Amed,

We're pleased to inform you that your manuscript has been judged scientifically suitable for publication and will be formally accepted for publication once it meets all outstanding technical requirements.

Within one week, you'll receive an e-mail detailing the required amendments. When these have been addressed, you'll receive a formal acceptance letter and your manuscript will be scheduled for publication.

An invoice for payment will follow shortly after the formal acceptance. To ensure an efficient process, please log into Editorial Manager at https://www.editorialmanager.com/pdig/ click the 'Update My Information' link at the top of the page, and double check that your user information is up-to-date. For billing related questions, please contact billing support at https://plos.my.site.com/s/.

Kind regards,

Haleh Ayatollahi

Section Editor

PLOS Digital Health

Additional Editor Comments (optional):

Reviewers' comments:

Reviewer's Responses to Questions

**Comments to the Author**

Reviewer #3: All comments have been addressed

publication criteria?

Reviewer #3: Yes

3. Has the statistical analysis been performed appropriately and rigorously?

Reviewer #3: Yes

4. Have the authors made all data underlying the findings in their manuscript fully available (please refer to the Data Availability Statement at the start of the manuscript PDF file)?

Reviewer #3: Yes

5. Is the manuscript presented in an intelligible fashion and written in standard English?

PLOS Digital Health does not copyedit accepted manuscripts, so the language in submitted articles must be clear, correct, and unambiguous. Any typographical or grammatical errors should be corrected at revision, so please note any specific errors here.

Reviewer #3: Yes

Reviewer #3: All the comments of the reviewer have been properly addressed and the manuscript is now ready for further processing.

**Do you want your identity to be public for this peer review?** For information about this choice, including consent withdrawal, please see our Privacy Policy

Reviewer #3: No
